# TabCaps: A Capsule Neural Network for Tabular Data Classification with BoW Routing

**Jintai Chen[1], Kuanlun Liao[1], Yanwen Fang[2], Danny Z. Chen[3], Jian Wu[*4]**

[1]College of Computer Science and Technology, Zhejiang University
[2]Department of Statistics & Actuarial Science, The University of Hong Kong
[3]Department of Computer Science and Engineering, University of Notre Dame
[4]Department of Public Health, Zhejiang University School of Medicine

## Abstract

Records in a table are represented by a collection of heterogeneous scalar features. Previous work often made predictions for records in a paradigm that processed each feature as an operating unit, which requires to well cope with the heterogeneity. In this paper, we propose to encapsulate all feature values of a record into vectorial features and process them collectively rather than have to deal with individual ones, which directly captures the representations at the data level and benefits robust performances. Specifically, we adopt the concept of *capsules* to organize features into vectorial features, and devise a novel capsule neural network called TabCaps to process the vectorial features for classification. In TabCaps, a record is encoded into several vectorial features by some optimizable multivariate Gaussian kernels in the primary capsule layer, where each vectorial feature represents a specific *profile* of the input record and is transformed into senior capsule layer under the guidance of a new straightforward routing algorithm. The design of routing algorithm is motivated by the Bag-of-Words (BoW) model, which performs capsule feature grouping straightforwardly and efficiently, in lieu of the computationally complex clustering of previous routing algorithms. Comprehensive experiments show that TabCaps achieves competitive and robust performances in tabular data classification tasks. Codes are available at `https://github.com/WhatAShot/TabCaps`.

## 1 Introduction

Tabular data are ubiquitous in real world applications, which records abundantly meaningful information such as medical examination results (Hassan et al., 2020) and company financial statements (Addo et al., 2018). Previous methods often processed a record by treating the scalar feature values as the operating units. For example, decision tree based methods (Breiman et al., 1984; Chen & Guestrin, 2016) used one tabular feature in each decision step, and neural networks (Gorishniy et al., 2021; Chen et al., 2022) elaborately executed feature-wise interactions to capture higher-level semantics.

However, it is intractable to design effective feature-wise interaction approaches (Grinsztajn et al., 2022; Ng, 2004) due to the heterogeneity among features. In this paper, we propose a novel paradigm for supervised tabular learning, which encapsulates all feature values of records into vectorial features and directly conducts on the vectorial feature level. Such design utilizes the sufficient representation space of the vectorial feature format to probably learn the comprehensive data level semantics, and avoids executing complex interactions among heterogeneous features.

To this end, we borrow the concept of *capsules* (Sabour et al., 2017) to organize vectorial features, and propose a novel capsule neural network (CapsNet) called TabCaps for tabular data classification. In TabCaps, several optimizable multivariate Gaussian kernels encode all the features of each record into the primary capsules, in which features in a vector format represent the marginal likelihoods of the record in reference to the corresponding multivariate Gaussian distributions. We set the scale and location parameters of these Gaussian kernels learnable, thus allowing these kernels to model some plausible *data patterns* for the dataset and each primary capsule feature represents a specific *profile*

---

∗ Corresponding author. E-mail: `wujian2000@zju.edu.cn`.

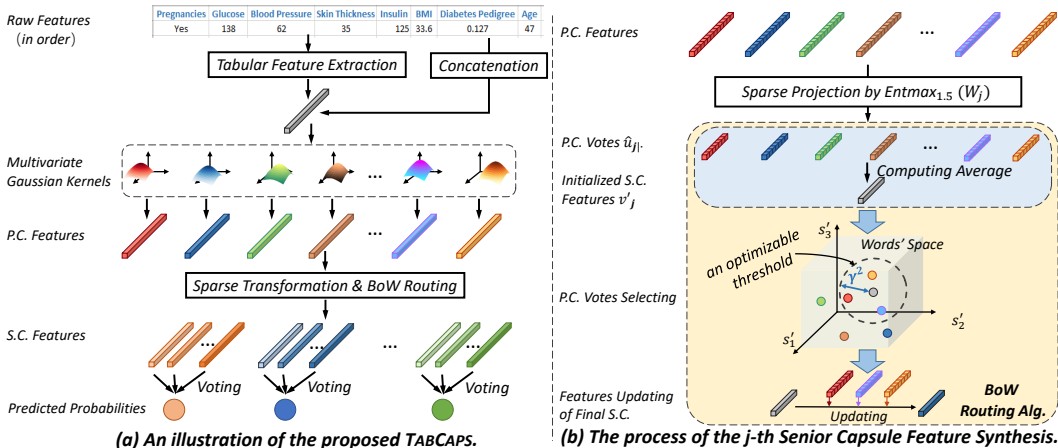

Figure 1: **Illustrating *(a)* our proposed TABCAPS (without the decoder) and *(b)* the process of senior capsule feature synthesis from features in primary capsules (taking features in the $j$-th senior capsule as example).** "P.C" denotes "primary capsule" and "S.C." denotes "senior capsule".

of the input record that measures the likelihood to these *data patterns*. Unlike previous CapsNets that used one senior capsule to predict the belonging probability for one class, we allot multiple senior capsules for a class, motivated by ensemble learning. In previous CapsNets, primary capsule features were transformed into senior capsules by an affinity projection and a routing algorithm (Sabour et al., 2017; Hinton et al., 2018) that groups similar primary capsule features by clustering processes. In TABCAPS, a novel sparse projection method and a novel straightforward routing algorithm are proposed to perform the feature transformation from primary to senior capsules.

Our proposed routing algorithm is much more efficient than previous routing algorithms, since the primary capsules in previous CapsNets captured some unknown semantics from unstructured data (e.g., images) and had to apply feature clustering in an iterative process to attain higher-level semantics. In our TABCAPS, features in primary capsules are the likelihood w.r.t. the Gaussian distributions, which represents stable semantics and allows to perform a straightforward routing algorithm. Motivated by the bag-of-words (BoW) model (Salton & Lesk, 1965) which is efficient in similar information search (Pineda et al., 2011), we propose a straightforward routing (called *BoW Routing*) for TABCAPS. The proposed *BoW Routing* computes similarities between primary capsule features and the initialized senior capsule features by counting the co-occurrences of "words" (implemented to be learnable templates), and incorporates those primary capsule features whose similarities are beyond an adaptive threshold to update senior capsule features. Similar to previous CapsNets, we also present a tabular-data-suited decoder, which is tailored for tabular feature reconstruction tasks (e.g., missing value imputation).

**Contributions**. (i) For the first time, we propose to encapsulate all the features of a record in table into vectorial features as operating units, which avoids inefficient interactions between heterogeneous tabular features and directly learns the data-level semantics. (ii) We propose a CapsNet tailored for supervised tabular learning, in which we devise a new type of primary capsule with learnable multivariate Gaussian kernels and conduct a novel straightforward routing to synthesize senior capsule features at low costs. (iii) Experiments on real-world datasets validate that TABCAPS attain robust performances in tabular data classification.

## 2 RELATED WORK

**Capsule Neural Networks.** Capsule neural networks (Ribeiro et al., 2022) were first proposed to deal with image data, where a capsule represents an object whole or object part in a vector or matrix format. Firstly, primary capsules capture the basic object parts by using templates (Kosiorek et al., 2019), optical flow analysis (Sabour et al., 2021), or location-based operations (Hinton et al., 2018). Features in primary capsules are linearly projected to transpose the poses of object parts, so as to synthesize senior capsule features (representing object wholes) guided by a routing algorithm. Most routing algorithms performed iteration processes at high computational costs (Sabour et al., 2017; Hinton et al., 2018), since the primary capsules handle different semantics in dealing with different samples. To avoid iterations, some straightforward approaches were proposed (Choi et al., 2019; Ahmed & Torresani, 2019; Chen et al., 2021; Ribeiro et al., 2020). However, these straightforward

approaches were for image tasks and some of them got rid of the part-to-whole learning assumption. At present, CapsNets have been widely used for image processing (Rajasegaran et al., 2019; Hinton et al., 2018; Chen et al., 2021), point clouds (Zhao et al., 2019b; 2020), recommendation in shopping (Li et al., 2019), and language tasks (Zhao et al., 2019a). However, to our best knowledge, no bespoke CapsNets were proposed for general supervised tabular learning tasks. This paper presents a novel CapsNet for tabular data, in which primary capsules learn some specific kinds of *data patterns* for the dataset rather than the data-adaptive "object parts".

**Supervised Tabular Learning.**   Various methods were proposed for tabular data processing, mainly including decision tree algorithms (Breiman et al., 1984; Quinlan, 1979; Chen & Guestrin, 2016; Prokhorenkova et al., 2018; Ke et al., 2017) and neural network (NN) based methods (Arik & Pfister, 2020; Popov et al., 2019; Chen et al., 2022; Abutbul et al., 2021). Different from tree based methods that split tabular feature spaces by some information metrics (Quinlan, 1979; 2014; Breiman et al., 1984) and sought for possible decision paths, NN based methods often introduced attention and grouping mechanisms (Chen et al., 2022; Arik & Pfister, 2020; Guo et al., 2017; Wang et al., 2017; Gorishniy et al., 2021; Yan et al., 2023) for tabular feature interactions to yield high-level features. However, tabular features in a table are heterogeneous and are relation-agnostic, and thus it is hard to conduct all the feature interactions in the right way.

## 3 METHODOLOGY

We propose to classify each record in a table by processing a vectorial feature containing all its feature values in order. See the pipeline of our TABCAPS in Fig. 1(a), to better depict an input record, its raw features are first processed by a simple function (e.g., an *Abstract Layer* (Chen et al., 2022) or a linear projection layer) to yield more comprehensive features, which are then concatenated with raw features and we obtain the basal vectorial feature of size $m$, $x \in \mathbb{R}^m$. Then, the vectorial feature is projected into primary capsules (in a vector format) by various learnable multivariate Gaussian kernels. Each vectorial feature (in a primary capsule) is transformed into *feature votes* for senior capsules by the proposed sparse projection (see Sec. 3.2.1), and senior capsule features are finally obtained by selecting and merging *feature votes*, guided by our *BoW Routing* (see Sec. 3.2.2).

### 3.1 HOW TO ENCODE RECORDS INTO PRIMARY CAPSULES?

Unlike the known CapsNets for images that utilized various primary capsules to represent different parts of objects, in TABCAPS, different primary capsules represent various *profiles* for just one record. Formally, a record represented by a vectorial feature $x \in \mathbb{R}^m$ is transformed into a feature $u_i \in \mathbb{R}^m$ for the $i$-th primary capsule by a multivariate Gaussian kernel $k_i$, as:

$$u_i = k_i(x; \mu_i, \Sigma_i) = \frac{\exp(-\frac{1}{2}(x - \mu_i)^T \Sigma_i^{-1}(x - \mu_i))}{(2\pi)^{m/2}|\Sigma_i|}, \tag{1}$$

where the location parameter vector $\mu_i \in \mathbb{R}^m$ and the scale parameters $\Sigma_i = \mathrm{diag}(\sigma_1, \sigma_2, \ldots, \sigma_m)$ of the multivariate Gaussian kernel $k_i$ are set to be learnable, so that TABCAPS allows to use proper multivariate Gaussian kernels in different datasets. Since features in a table are typically regarded as independent, we set $\Sigma$ as a diagonal matrix. After Eq. (1), a layer normalization is performed on each $u_i$ ($i = 1, 2, \ldots, N_p$, $N_p$ is the count of the primary capsules) for normalization following Tsai et al. (2019). Specially, we also encapsulate the original feature values (denoted by $x$) to be an additional capsule feature.

This design can be interpreted that a primary capsule models an *data pattern* by determining the parameters $(\mu, \Sigma)$ of the multivariate Gaussian kernels, and various primary capsule features of an input record are represented as the likelihoods in reference to these *data patterns*. It is beneficial to find some dataset-specific *data patterns*, which allows to implicitly model the tabular feature relations.

### 3.2 HOW TO SYNTHESIZE SENIOR CAPSULE FEATURES?

Motivated by ensemble learning, we use multiple senior capsules to learn the semantics for one target class and obtain the final prediction by voting, which also meets the need of irregular target pattern fitting (Grinsztajn et al., 2022). To attain the senior capsule features, primary capsule features are transformed by a sparse weight matrix into *feature votes*, and a newly proposed *BoW Routing* selects *feature votes* to synthesize senior capsule features. Note that the elements in vectorial features are linearly transformed (reversible), and the non-linear mapping (the routing) is performed by taking vectorial features as operating units.

### 3.2.1 SPARSE PROJECTION

The affinity projection in previous work (Sabour et al., 2017) uses a learnable weight $W_{j|i} \in \mathbb{R}^{m \times n}$ to linearly map the $i$-th primary capsule feature $u_i \in \mathbb{R}^m$ to yield a *feature vote* $\hat{u}_{j|i} \in \mathbb{R}^n$ for the $j$-th senior capsule, by $\hat{u}_{j|i} = u_i W_{j|i}$. We modify the weight matrix by adding a sparse operation *entmax* (Peters et al., 2019), and the transformation is computed by:

$$\hat{u}_{j|i} = u_i W'_{j|i} \, , \; W'_{j|i} = entmax_\alpha(W_{j|i}), \tag{2}$$

where $\alpha = 1.5$ as default. In this way, the weight matrix $W'_{j|i}$ shall be sparse and only parts of elements of $u_{j|i}$ is used to synthesize the *feature votes* and the target-irrelevant elements are excluded. Different from sparse attention operations, the *entmax* is applied to the weight matrix. Thus, the capsule feature component selection is consistent for different records.

### 3.2.2 BoW ROUTING

Given the synthesized *feature votes* $\hat{u}_{j|i}$ for the $j$-th senior capsule (where $i = 1, 2, \ldots, N_p$, $j = 1, 2, \ldots, N_s$, $N_p$ and $N_s$ are the amounts of primary capsules and senior capsules respectively), we use a routing algorithm to synthesize senior capsule features by selecting and merging $\hat{u}_{j|i}$. Note that the vanilla routing algorithm was a clustering-like approach (Sabour et al., 2017), because a primary capsule may capture any kinds of object parts from images and thus the "cluster centroids" are unknown before the routing step. In our model, each primary capsule learns concrete semantics that represent record *profiles* in reference to a specific multivariate Gaussian distribution, making it possible to predict the initialized senior capsule features ("cluster centroids") directly using a straightforward function w.r.t. semantics (features) captured by primary capsules. Here we simply use the average of the *feature votes* $\hat{u}_{j|i}$ to initialize senior capsule features $v'_j$ by $v'_j = \sum_i^{N_p} \hat{u}_{j|i}/N_p$.

To select some *feature votes* for senior capsule feature synthesis, we measure the similarities between the initialized senior capsule feature $v'_j$ and the primary capsule *feature vote* $\hat{u}_{j|i}$. An intuitive solution is to compute the similarity directly. However, directly computing similarity might hinder the performance, which overlooks the special data patterns and importance differences of different vectorial feature elements. Motivated by the success of the bag-of-words (BoW) model in similarity measure for information retrieval, we introduce the BoW model to the routing algorithm. A classical BoW model first clusters the given features, and collects a word dictionary whose words are the centroids of feature clusters. Then, a feature can be quantized as the "occurrence" of various words, and feature similarities can be computed by counting the co-occurring words. In this way, the similarity is computed in the semantic space guided by the "words" in discrete steps. Our differentiable *BoW Routing* is depicted as follow.

**Occurrence status computing.** The dictionary words in the traditional BoW model are computed according to the extracted features. Motivated by Chen et al. (2021), we directly implement the word dictionary with a learnable parameter set of size $M$, $\{s_m \mid m = 1, 2, \ldots, M\}$, with each "word" $s_m \in \mathbb{R}^n$ (of the identical size as the *feature votes* $\hat{u}_{j|i}$). To compute the "occurrences" of the "words", an occurrence status (a scalar) between $\hat{u}_{j|i}$ and the initialized senior capsule feature $v'_j$ for the "word" $s_m$ is defined by:

$$p_{j|i;m} = \text{sigmoid}(\hat{u}_{j|i}^T \cdot s'_m) \in (0, 1), \; p_{j;m} = \text{sigmoid}(v'^T_j \cdot s'_m) \in (0, 1), \tag{3}$$

where $p_{j|i;m}$ and $p_{j;m}$ represent the degrees (i.e., the occurrence statuses) in which $u_{j|i}^T$ and $v'^T_j$ reveal the semantics of $s_m$, we let $\mathbf{p}_{j|i} = [p_{j|i;1}, p_{j|i;2}, \ldots, p_{j|i;M}]$ and $\mathbf{p}_j = [p_{j;1}, p_{j;2}, \ldots, p_{j;M}]$ represent semantic codes of $\hat{u}_{j|i}$ and $v'_j$. In computing Eq. (3), the "word" $s_m$ is normalized into $s'_m$ by the $l_2$ norm to avoid the impact of the vector lengths. In traditional BoW models, an occurrence status is represented as binary (0 or 1), but by our routing algorithm, we generalize the discrete binary status to a continuous status form (in $(0, 1)$) using the "sigmoid" function.

**Feature vote merging.** To compute the similarity between the occurrence statuses $\mathbf{p}_{j|i}$ and $\mathbf{p}_j$, traditional BoW models may use the Jaccard distance. Since the occurrence statuses are continuous values in our routing algorithm, to which the Jaccard distance cannot be simply applied, we compute the pairwise distances $d_{j|i}$ between $\mathbf{p}_{j|i}$ and $\mathbf{p}_j$ using the squared Euclidean distance function $D$:

$$d_{j|i} = D(\mathbf{p}_{j|i}, \mathbf{p}_j) = \sum_{m=1}^M (p_{j|i;m} - p_{j;m})^2. \tag{4}$$

Note that the squared Euclidean distance has been proved to be an effective approximation of the Jaccard distance (Reina et al., 2014). Based on the pairwise distances $d_{j|i}$ of *feature votes* and initialized senior capsule features, we can directly obtain the neighboring *feature votes* $\hat{u}_{j|i}$ to update the features in the $j$-th senior capsule by performing $k$-Nearest Neighbor ($k$-NN) search. However, we find that $k$-NN can lead to instability in training, since the distributions of *feature votes* are unknown and $k$-NN might select some *feature votes* with extreme values.

To gain training stability, we propose to use an adaptive threshold to determine a range around the initialized senior capsule feature $v'_j$ (see Fig. 1(b)). We set this threshold as a learnable positive value $\gamma^2 > 0$, and the selection of *feature votes* is determined by:

$$q_{j|i} = \varepsilon(\gamma^2 - d_{j|i}), \tag{5}$$

where $\varepsilon(\cdot)$ is the Heaviside function, and if and only if $q_{j|i} = 1$, it indicates that the *feature vote* from the $i$-th primary capsule, $\hat{u}_{j|i}$, is selected. Then, we compute the weights for the *feature votes* and obtain the final senior capsule feature $v_j$ by a weighted sum, as:

$$w_{j|i} = \text{softmax}(q_{j|i} \cdot e^{-d_{j|i}}), \ v_j = \sum_i^{N_p} w_{j|i}\hat{u}_{j|i}, \tag{6}$$

where the "softmax" operation guarantees all $w_{j|i} > 0$ (giving small weights to unselected *votes*), avoiding possible failure if no *feature vote* is selected to yield $v_j$. In this way, our proposed *BoW Routing* performs a straightforward process to obtain the senior capsule features.

**Remarks.** Distinct from previous straightforward routing algorithms (Chen et al., 2021; Ribeiro et al., 2020) for images, the efficiency of our *BoW Routing* partially results from the innovative definition of the primary capsules that learns concrete semantics, allowing to predict initialized senior capsule features directly, in lieu of learning unknown object parts and requiring clustering-like routing algorithms to group them. Our *BoW Routing* algorithm, with its strong ability to extract semantic patterns and flexible measurement of similarity, allows us to utilize a straightforward average method for initializing the senior capsule features.

### 3.3 SENIOR CAPSULE VOTING FOR CLASSIFICATION

Similar to the known CapsNets, we let the vector lengths of the senior capsule features represent the existence probabilities of one semantics, and this work models each target class by multiple senior capsules motivated by ensemble learning. Suppose there are $K$ classes. We divide all the senior capsules into $K$ groups, such that the features in each group vote for one particular class. The classification outcomes are defined by:

$$\mathbf{c} = \text{softmax}([l_1, \ldots, l_k, \ldots, l_K]), \ \hat{y} = \text{argmax}(\mathbf{c}), \tag{7}$$

where the vector $\mathbf{c}$ represents the predicted probabilities for all the $K$ classes, and $\hat{y}$ denotes the predicted label. A value $l_k$ in Eq.(7) denotes the mean of the senior capsule feature vector lengths for the $k$-th class (i.e., in the group $G_k$), which is defined by: $l_k = \sum_{j \in G_k} ||v_j||_2 / ||G_k||$. Note that we dropout 20% senior capsules in a group before voting in training. We use the margin loss in training our TABCAPS following Sabour et al. (2017), as:

$$\mathcal{L} = \sum_{k=1}^{K} T_k \max(0, t^+ - l_k)^2 + \lambda(1 - T_k)\max(0, l_k - t^-)^2, \tag{8}$$

where $T_k = 1$ if the $k$-th class is the target, and otherwise $T_k = 0$. We set $t^+ = 0.9, t^- = 0.1$, and $\lambda = 0.5$ following Sabour et al. (2017).

### 3.4 A DECODER FOR TABULAR FEATURE RECONSTRUCTION

We build a decoder based on the *Abstract Layers* (Chen et al., 2022) for tabular feature reconstruction. We stack two *Abstract Layers* in sequence with a shortcut performing feature concatenation, as shown in Fig. 2. Finally, a fully connected layer is on top of the decoder to produce the reconstructed tabular features. Since there are multiple senior capsules voting for a target class, the decoder input is the concatenation of features in the group of senior capsules that votes for the correct class. We do not use the decoder as regularization like previous CapsNets (Sabour et al., 2017) and TABCAPS without the decoder can perform well on classification tasks (see Fig. 3). The decoder is used when we attempt to reconstruct raw features for some tasks (e.g., missing value imputation), which is trained under the specification of the mean square error (MSE) that is scaled down by $5 \times 10^{-6}$ and jointly works with the loss $\mathcal{L}$ (in Eq. (8)).

## 4 EXPERIMENTS

### 4.1 EXPERIMENTAL SETUPS

**Datasets.** We conduct experiments on 8 real-world tabular datasets, including *Heart Failure Prediction (Heart)* (Chicco & Jurman, 2020), *Click* (Cup, 2012), *Diabetes* (Ukani, 2020), *EEG Eye State (EEG)* (Dua & Graff, 2015), *Gas Concentrations (Gas)* (Vergara et al., 2012), *Hill-Valley (Hill)* (Lee & Franz, 2008), *Higgs-small (Higgs)* (Baldi et al., 2014) and *Epsilon* (Challenge, 2008). All of these datasets have few or no categorical features, eliminating the impacts of categorical feature embedding approaches in model comparison. In pre-processing, data are standardized by the *z*-score normalization. We follow the train-test split and the data pre-processing in (Popov et al., 2019) for the *Click* dataset. For the other datasets, we randomly divide 80% of the data into the training sets, and the rest 20% are for test. We further sample 15% training data for validation. Experiments were run for 5 times with regard to random seeds.

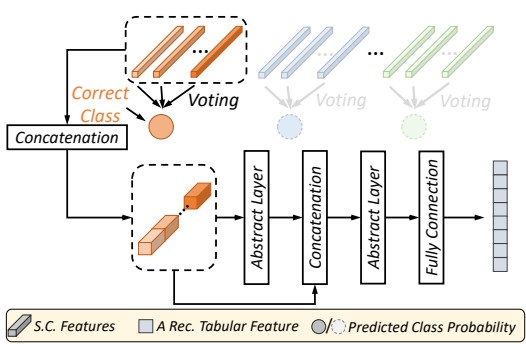

Figure 2: **The decoder for reconstructing raw tabular features.** The decoder input is the senior capsule features in the group that votes for the correct class. "Rec." = "Reconstructed" and "S.C." = "Senior Capsule".

**Implementation details and comparison baselines.** Our method is implemented with PyTorch on Python 3.7, and is run on GeForce RTX 2080 Ti. In training, we set the batch size as 2,048. We use the QHAdam optimizer (Ma & Yarats, 2019) with weight decay rate $10^{-5}$, $v = (0.7, 0.99)$, and $\beta = (0.95, 0.998)$. We compare TABCAPS with several baseline models, including TabNet (Arik & Pfister, 2020), NODE (Popov et al., 2019), FT-Transformer (Gorishniy et al., 2021), DANet-24 (Chen et al., 2022), Net-DNF (Abutbul et al., 2021), XGboost (Chen & Guestrin, 2016), Cat-Boost (Prokhorenkova et al., 2018), the Vector CapsNet (Sabour et al., 2017), and fully connected neural networks (FCNNs) (Nair & Hinton, 2010) with lasso and mixup (Zhang et al., 2018) ($\alpha = 1.0$) respectively. The Vector CapsNet (Sabour et al., 2017) was designed for images; we replace its convolution layer by a fully connected layer for tabular data. Following Popov et al. (2019); Arik & Pfister (2020), the performances of all the methods are hyperparameter-tuned on the validation sets (e.g., by grid search). The hyperparameter search spaces and search algorithms are given in Appendix A.1. We employ the "log-loss" metric to measure the model performances, which is considered as a more sensitive metric than accuracy, and is widely used (e.g., in Kaggle competitions). All the performances of the compared methods are obtained by running official codes.

### 4.2 CLASSIFICATION PERFORMANCE COMPARISON

The classification performances are reported in Table 1. One can see that our TABCAPS obtains the best or second best performances on 7 out of 8 datasets, and is especially superior on some datasets by clear margins. In addition, we also report the computational complexity and efficiency on the *Diabetes* dataset, using the metrics of model size (# param.) and fps (the amount of records that a model processes per second). Since XGboost and CatBoost are traditional machine learning models, we do not report their complexity. One can see that TABCAPS is much more efficient and greatly lighter (over $10\times$) than the other state-of-the-art methods (e.g., FT-Transformer, Net-DNF, and DANet-24), and significantly outperforms the models with similar model complexity (e.g., regularized FCNNs and Vector CapsNet). Also, the results compared with Vector CapsNet shows the superiority of TABCAPS, obtaining better performances with lower complexity on tabular data.

### 4.3 EXTREME GENERALIZATION TEST

To further examine the generalization capability of TABCAPS, we attempt to test it with train-test split that the training data and test data are with very dissimilar feature values. We conduct t-SNE projections (with default hyper-parameters of the "sklearn" package [1]) on all the records of datasets,

---

[1] https://scikit-learn.org/stable/modules/generated/sklearn.manifold.TSNE.html

Table 1: **Classification Performances.** The best and second best performances of deep learning approaches are respectively marked in **bold** and underlined. Note that the reported log-loss values (the lower the better) are with a $100\times$ factor. The model size (# param.) and inference speed (fps) are on the *Diabetes* dataset. The performances are reported as "mean$_{\pm\text{std}}$".

| Method | Click | Diabetes | EEG | Gas | Heart | Hill | Higgs | Epsilon | # param. | fps |
|---|---|---|---|---|---|---|---|---|---|---|
| XGboost | $62.253_{\pm0.02}$ | $14.338_{\pm0.03}$ | $14.117_{\pm0.02}$ | $2.087_{\pm0.06}$ | $32.371_{\pm0.04}$ | $69.049_{\pm1e\text{-}3}$ | $53.158_{\pm0.01}$ | $26.748_{\pm1e\text{-}3}$ | – | – |
| Catboost | $64.273_{\pm0.08}$ | $14.777_{\pm0.07}$ | $18.423_{\pm0.12}$ | $2.064_{\pm0.05}$ | $30.043_{\pm0.14}$ | $69.174_{\pm0.006}$ | $53.273_{\pm0.05}$ | $27.228_{\pm2e\text{-}3}$ | – | – |
| TabNet | $\underline{62.303}_{\pm0.03}$ | $17.964_{\pm0.04}$ | $45.340_{\pm0.04}$ | $4.647_{\pm0.04}$ | $44.967_{\pm0.01}$ | $87.804_{\pm0.07}$ | $54.668_{\pm0.03}$ | $26.743_{\pm0.02}$ | 3.4M | 73.1 |
| Net-DNF | $67.633_{\pm0.02}$ | $13.767_{\pm0.02}$ | $17.386_{\pm0.01}$ | $\mathbf{1.229}_{\pm0.04}$ | $55.371_{\pm0.03}$ | $\underline{15.787}_{\pm0.03}$ | $53.417_{\pm0.02}$ | $27.122_{\pm0.03}$ | 8.5M | 175.2 |
| NODE | $63.206_{\pm0.05}$ | $45.951_{\pm0.03}$ | $47.654_{\pm0.04}$ | $38.774_{\pm0.04}$ | $46.541_{\pm0.04}$ | $69.220_{\pm0.04}$ | $61.864_{\pm0.08}$ | $27.838_{\pm0.54}$ | 13.4M | 145.2 |
| FT-Transformer | $70.487_{\pm0.02}$ | $\underline{12.382}_{\pm0.03}$ | $\mathbf{7.446}_{\pm0.06}$ | $2.258_{\pm0.04}$ | $\mathbf{27.547}_{\pm0.05}$ | $20.084_{\pm0.03}$ | $\underline{53.310}_{\pm0.02}$ | $\underline{25.958}_{\pm0.85}$ | 9.3M | 284.7 |
| DANet-24 | $73.708_{\pm0.02}$ | $13.338_{\pm0.02}$ | $9.301_{\pm0.04}$ | $2.171_{\pm0.02}$ | $49.643_{\pm0.04}$ | $24.763_{\pm0.03}$ | $\mathbf{53.033}_{\pm0.01}$ | $26.431_{\pm0.01}$ | 5.5M | 54.9 |
| FCNN w/ mixup | $63.863_{\pm0.07}$ | $12.715_{\pm0.05}$ | $9.572_{\pm0.07}$ | $2.083_{\pm0.06}$ | $36.742_{\pm0.02}$ | $56.005_{\pm0.05}$ | $56.787_{\pm0.04}$ | $27.467_{\pm0.03}$ | 0.7M | 594.3 |
| FCNN w/ lasso | $87.005_{\pm0.17}$ | $41.071_{\pm0.75}$ | $31.852_{\pm0.05}$ | $4.141_{\pm0.06}$ | $44.881_{\pm0.06}$ | $69.302_{\pm0.01}$ | $132.102_{\pm0.07}$ | $32.282_{\pm0.02}$ | 0.7M | 568.8 |
| Vector CapsNet | $64.135_{\pm0.05}$ | $52.635_{\pm0.03}$ | $53.587_{\pm0.06}$ | $161.547_{\pm0.03}$ | $58.516_{\pm0.04}$ | $51.591_{\pm0.02}$ | $62.654_{\pm0.02}$ | $54.252_{\pm0.02}$ | 0.4M | 318.5 |
| TABCAPS (Ours) | $\mathbf{62.054}_{\pm0.04}$ | $\mathbf{12.043}_{\pm0.03}$ | $\underline{8.130}_{\pm0.05}$ | $\underline{2.013}_{\pm0.03}$ | $\underline{34.047}_{\pm0.02}$ | $\mathbf{14.301}_{\pm0.04}$ | $53.776_{\pm0.03}$ | $\mathbf{25.821}_{\pm0.02}$ | 0.2M | 501.1 |

Table 2: **Extreme generalization performances.** The best and second best performances of deep learning approaches are respectively marked in **bold** and underlined. Note that the reported log-loss values (the lower the better) are with a $100\times$ factor. The *Epsilon* dataset is not included due to its extremely high computation complexity in conducting t-SNE projection.

| Method | Click | Diabetes | EEG | Gas | Heart | Hill | Higgs |
|---|---|---|---|---|---|---|---|
| Training-Test Split | | | | | | | |
| XGboost | $66.070_{\pm0.03}$ | $65.886_{\pm0.09}$ | $70.654_{\pm0.02}$ | $31.504_{\pm0.04}$ | $35.650_{\pm0.01}$ | $69.657_{\pm0.09}$ | $54.557_{\pm0.04}$ |
| Catboost | $63.925_{\pm0.04}$ | $68.819_{\pm0.06}$ | $68.799_{\pm0.04}$ | $18.864_{\pm0.04}$ | $35.207_{\pm0.08}$ | $69.162_{\pm0.03}$ | $54.632_{\pm0.07}$ |
| TabNet | $115.907_{\pm0.11}$ | $225.22_{\pm0.08}$ | $79.666_{\pm0.07}$ | $158.618_{\pm0.03}$ | $44.967_{\pm0.06}$ | $89.114_{\pm0.08}$ | $55.763_{\pm0.11}$ |
| Net-DNF | $67.625_{\pm0.02}$ | $\underline{58.792}_{\pm0.05}$ | $68.261_{\pm0.04}$ | $15.124_{\pm0.03}$ | $55.371_{\pm0.07}$ | $48.301_{\pm0.04}$ | $55.738_{\pm0.06}$ |
| NODE | $\underline{63.839}_{\pm0.04}$ | $67.021_{\pm0.04}$ | $68.357_{\pm0.04}$ | $57.698_{\pm0.06}$ | $46.541_{\pm0.03}$ | $69.771_{\pm0.10}$ | $61.870_{\pm0.03}$ |
| FT-Transformer | $78.431_{\pm0.11}$ | $59.283_{\pm0.04}$ | $68.278_{\pm0.07}$ | $\mathbf{6.416}_{\pm0.06}$ | $\mathbf{26.132}_{\pm0.05}$ | $66.972_{\pm0.05}$ | $\mathbf{53.970}_{\pm0.10}$ |
| DANet-24 | $74.401_{\pm0.02}$ | $59.736_{\pm0.06}$ | $69.021_{\pm0.03}$ | $10.395_{\pm0.01}$ | $49.643_{\pm0.04}$ | $\underline{37.976}_{\pm0.04}$ | $\underline{54.182}_{\pm0.01}$ |
| FCNN mixup | $66.052_{\pm0.05}$ | $60.262_{\pm0.04}$ | $68.850_{\pm0.08}$ | $25.102_{\pm0.04}$ | $35.674_{\pm0.17}$ | $67.126_{\pm1e\text{-}3}$ | $55.847_{\pm0.01}$ |
| FCNN lasso | $106.123_{\pm3e\text{-}3}$ | $67.082_{\pm0.04}$ | $93.170_{\pm0.04}$ | $61.310_{\pm0.02}$ | $76.854_{\pm0.03}$ | $75.853_{\pm0.02}$ | $106.580_{\pm0.06}$ |
| Vector CapsNet | $64.724_{\pm0.05}$ | $66.009_{\pm0.02}$ | $\underline{67.845}_{\pm0.04}$ | $163.193_{\pm0.04}$ | $60.848_{\pm0.04}$ | $64.743_{\pm0.09}$ | $62.791_{\pm0.02}$ |
| TABCAPS (Ours) | $\mathbf{63.355}_{\pm0.04}$ | $\mathbf{58.409}_{\pm0.02}$ | $\mathbf{67.471}_{\pm0.01}$ | $\underline{8.750}_{\pm0.06}$ | $\underline{34.503}_{\pm0.05}$ | $\mathbf{17.887}_{\pm0.04}$ | $54.707_{\pm0.07}$ |

and make the train-test split based on the t-SNE embedding (see the 2nd row in Table 2). We use the records inside the black rectangles for training (containing $\sim$70% data) and the rest for test. Notably, the rectangles are biasedly placed and the records for training and test do not overlap in the t-SNE embedding, so that they are clearly in different distributions. The performances reported in Table 2 show that our TABCAPS obtains best or second best performances among deep learning approaches on most of datasets. Such results imply that our method can learn robust features and generalize well. Besides, our TABCAPS exhibits considerably better generalization performances than the Vector CapsNet with very clear margins (e.g., TABCAPS obtains $8.750 \times 10^{-2}$ on *Gas* dataset, while the Vector CapsNet obtains $163.193 \times 10^{-2}$).

## 4.4 PERFORMANCE STABILITY DURING TRAINING

Many previous works required the early stop (Yao et al., 2007) to avoid over-fitting because they managed to exploit the potential of each tabular feature, and thus might be impacted by the uninformative features. However, our TABCAPS does not seem to require such training trick since it learns the data-level semantics directly. To show this, we compare the **real-time validation performances** (on log-loss values) during traning, among several regularization strategies (e.g., FCNN with lasso, mixup (Zhang et al., 2018)), commonly used neural networks (e.g., TabNet with sparse regularization, NODE and Net-DNF with ensemble learning strategies), and traditional boosting methods (XGboost and CatBoost), on the *Click* dataset. As shown in Fig. 3, most of these methods obtain their lowest log-loss values on the validation set after few training steps, and then their models gradually collapse as the training step increases. Notably, during the training process, all the corresponding loss values on training data keeps on decreasing or slightly fluctuating. One can see that the lasso and mixup

Table 3: **Ablation study results.** For the missing value imputation (MVI) test, we report the performances of a XGBoost model trained on training sets with filled values. The log-loss values are $100\times$.

| Method | | Performances | |
|---|---|---|---|
| Routing & projection | | Classification | fps |
| (1) *BoW Routing* & sparse projection | | 12.043±0.03 | 1666 |
| (2) dynamic routing & sparse projection | | 14.543±0.03 | 1342 |
| (3) *BoW Routing* & simple projection | | 13.166±0.03 | 1683 |
| Routing | Decoder | MVI (log-loss ($100\times$)) | |
| (3) *BoW Routing* | Ours | 22.531±0.01 | |
| (4) *BoW Routing* | fully connection | 26.338±0.07 | |
| (5) Auto-Encoder with a classifier | | 25.765±6e-3 | |
| Capsule Layers | | Classification | |
| (6) 2 | | 12.043±0.03 | |
| (7) 3 | | 6.254±0.04 | |
| (8) 4 | | 11.138±0.03 | |
| Others | | Classification | |
| (9) w/ Tabular Feature Extraction | | 12.043±0.03 | |
| (10) w/o Tabular Feature Extraction | | 12.684±0.02 | |
| (11) S.C. ensemble | | 12.043±0.03 | |
| (12) single S.C. | | 12.515±0.05 | |

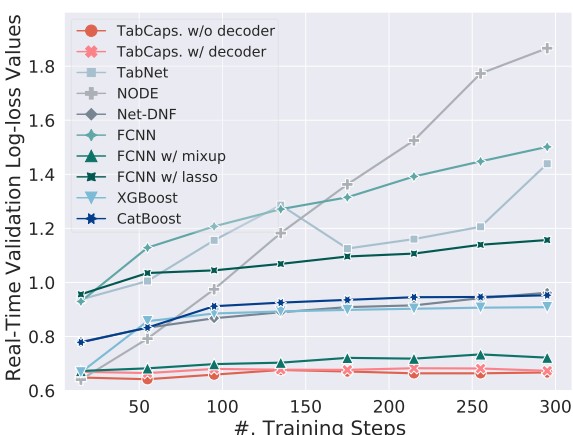

Figure 3: **Real-time validation performances (log-loss, the lower the better) on *Click* dataset.** A "step" (on the $x$-axis) indicates an "epoch" for neural networks and is a "boosting step" for XGboost and CatBoost.

regularization can alleviate this issues to some extent, but the validation log-loss values still increase slowly. Different from these methods, our TABCAPS with and without the decoder performs evidently stable, stabilizing the log-loss values at a low level (around $0.62$). This finding reveals an advantage of using vectorial features as operating units and suggests that our method can facilitate the users to obtain a well-performed model without intensely monitoring the performances on the validation sets or agonizing over the use of training tricks (e.g., regularization methods or early stop).

## 4.5 EFFECTS OF BOW ROUTING

Here we further inspect how our proposed *BoW Routing* determines the transformation paths for the vectorial features from primary capsules to senior capsules. We compute the averaged transformation weights ($w_{j|i}$ in Eq.(6)) over the whole test set and visualize them as in Fig. 4. One can see that a senior capsule features are yielded by merging limited vectorial *feature votes* from primary capsules. Besides, we found the weights are sparse and the values are similar to different samples (with the standard deviation values varying from $0.001$ to $0.003$), which suggests that primary capsules model some semantically meaningful *data patterns* and our *BoW Routing* succeeds in finding out clear decision paths for classification.

## 4.6 ABLATION STUDIES

We conduct ablation studies to examine the effects of the routing algorithms and the decoder (in a missing value imputation test) on the *Diabetes* dataset. We compare our *BoW Routing* with the dynamic routing (Sabour et al., 2017) on our architecture. See rows (1), (2), and (3) in Table 3. It is obvious that TABCAPS with our proposed *BoW Routing* outperforms that with the dynamic routing (Sabour et al., 2017) on the *Diabetes* data classification task by around $2.5 \times 10^{-2}$ on log-loss. We also compute the fps of the *BoW Routing* and dynamic routing, and find that *BoW Routing* is nearly $25\%$ faster than the dynamic routing, suggesting the superiority of our straightforwardness design. Compared with the simple projection (without *entmax*), our sparse projection is witnessed to be beneficial and is compatible with the *BoW Routing*, which shows that the use of *entmax* benefits in finding informative features and excludes the useless ones.

To inspect the capability of our proposed decoder, we conduct a test on missing value imputation, and employ an XGboost classifier to evaluate the effectiveness. We randomly mask out 20% of the feature values of the *Diabetes* training data, and simply fill the missing values with the means of feature values within the same classes. We train three models on the training set: (i) our TABCAPS with the proposed decoder (using the loss $\mathcal{L}$ in Eq. (8) and MSE), (ii) our TABCAPS with the fully-connection-based decoder of the Vector CapsNet (Sabour et al., 2017) (using $\mathcal{L}$ in Eq. (8) and MSE), and (iii) an auto-encoder with an auxiliary classifier that takes the hidden states as input (using MSE and cross-entropy loss). After training, we take the reconstructed feature values provided by these 3 decoders to refill the missing values respectively, and obtain 3 different training sets. To

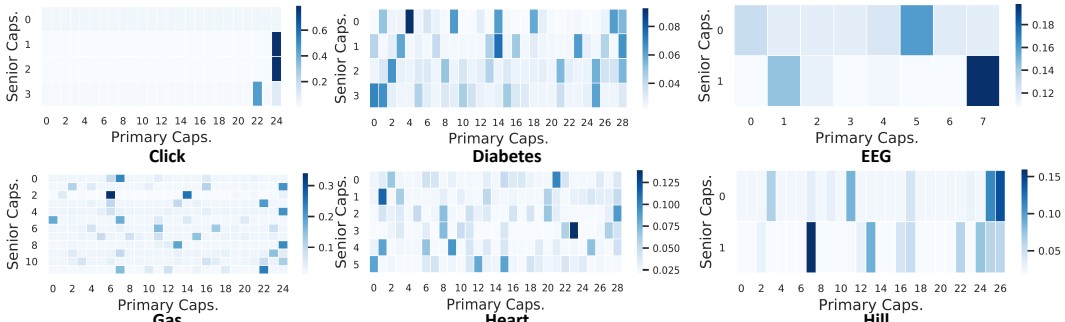

Figure 4: **The averaged transformation weights determined by *BoW Routing*.** The *x*-axis represents indices of primary capsules while the *y*-axis for senior capsules. The counts of primary capsules and senior capsules on different datasets are determined by the hyperparameter tuning.

fairly compare the qualities of the reconstructed values, we train 3 XGboost models with identical hyper-parameters on these 3 training sets, and report their performances on the original test set. A better performance indicates that the reconstructed values are more trusty.

As reported in Table 3, XGboost that is trained on the training set provided by our decoder is markedly superior to the other two methods, while the fully-connection-based decoder (Sabour et al., 2017) performs the worst. Notably, we train another XGboost on the training set in which the masked values remain missing, and find that this XGboost version still obtains 26.174 on the test set. Thus, one can see that our proposed decoder is able to yield significant performance gain in filling the missing values, while the fully-connection-based decoder predicts noisy values.

The effects of capsule layers were tested. See rows (6), (7), and (8). The 3-layer and 4-layer TABCAPS obtain better performances on the *Diabetes* dataset compared with the default 2-layer model. On the *Heart* dataset, similar conclusion is attained that TABCAPS with 2, 3, 4 capsule layers obtained log-loss values of 0.34047, 0.33992, and 0.33673, respectively. In general, we consider TABCAPS with different capsule layers perform competitive, and a 2-capsule-layer model is sufficient to attain a good performance. We also inspected the effects of the "tabular feature extraction" module at the bottom of TABCAPS (see Fig. 1), and the effects of the ensemble of the senior capsules. Comparing row (9) with (10), and row (11) with (12), it is evident that both of the tabular feature extraction and the senior capsule ensemble are beneficial, but are not the core reasons of the superior performances of TABCAPS. We consider it is the novel architecture makes TABCAPS perform well.

## 5    CONCLUSIONS AND FUTURE WORK

This paper proposed a novel and simple paradigm for supervised tabular learning, which encapsulated all features of a record into vectorial features as operating units and thus avoids interactions among heterogeneous tabular features. Consequently, our proposed new CapsNet (called TABCAPS), with a new primary capsule synthesis approach, a novel sparse projection, and a novel *BoW Routing*, coped with vectorial features and obtained superior performances on tabular data classification tasks at low computational costs. Specifically, the proposed *BoW Routing* performs straightforwardly without iterative clustering processes of previous CapsNets. This work is a beneficial attempt in the research areas of capsule neural networks, routing algorithms, and supervised tabular learning. Following previous works, our proposed method only considered classification tasks, and it would be interesting to explore how to use TABCAPS for tabular data regression. Moreover, it is also interesting to design other approach for directly learning data-level semantics without feature-wise interactions.

**Acknowledgements.** This research was partially supported by the National Key R&D Program of China under grant No. 2018AAA0102102 and National Natural Science Foundation of China under grants No. 62132017.

**Reproducibility Statement.**   To ensure the reproducibility, we include the settings of hyper-parameter tuning for various methods in Appendix A.1. The classification performances measured by "accuracy" and "ROC-AUC" are reported in Appendix A.2 for reference. Besides, an additional discussion between our proposed "multivariate Gaussian embedding" and "linear feature embedding" approaches are given in Appendix A.3.

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

# A APPENDIX

## A.1 HYPERPARAMETER-TUNING SETTINGS

As mentioned in the main paper, some hyperparameters of methods were turned for datasets by grid search or the HyperOpt library[2] with 50 steps of the Tree-structured Parzen Estimator (TPE), similar to the configurations in (Popov et al., 2019). The details are listed in the subsections below. As for XGBoost, CatBoost, FCNN, TabNet, DANets, Vector CapsNet, and our TABCAPS, we used the HyperOpt hyperparameter search library; as for NODE and Net-DNF, we used grid search (following the settings in their papers). For FT-Transformer (Gorishniy et al., 2021), we used their official open source.

The hyperparameter search spaces and search algorithms of XGBoost (Chen & Guestrin, 2016), CatBoost (Prokhorenkova et al., 2018), NODE (Popov et al., 2019), and FCNN (Nair & Hinton, 2010) were set identical as in (Popov et al., 2019). We set the maximal number of the trees in XGBoost and CatBoost up to 4096 (instead of 2048 used in NODE (Popov et al., 2019)), since we found that 2048 is not enough for XGBoost and CatBoost to obtain good performances. We also set the maximal layer number in FCNN up to 9. Besides, the hyperparameter search settings of Net-DNF (Abutbul et al., 2021) following its original paper. The architectures of FCNN with lasso regularization or mixup were constructed following the FCNN in (Popov et al., 2019). For both of DANet-24 (Chen et al., 2022) and TabNet (Arik & Pfister, 2020), we set the hyperparameter search spaces to contain all the possible values provided by the main papers.

### A.1.1 XGBOOST

For easy re-implementation, the hyperparameter variable names followed the sklearn package[3] settings.

- *eta (learning rate)*: Log-Uniform distribution $[e^{-7}, 1]$;
- *max_depth*: Discrete uniform distribution $[2, 10]$;
- *subsample*: Uniform distribution $[0.5, 1]$;
- *colsample_bytree*: Uniform distribution $[0.5, 1]$;
- *colsample_bylevel*: Uniform distribution $[0.5, 1]$;
- *min_child_weight*: Uniform distribution $[e^{-16}, e^5]$;
- *alpha*: Uniform choice $\{0$, Log-Uniform distribution $[e^{-16}, e^2]\}$;
- *lambda*: Uniform choice $\{0$, Log-Uniform distribution $[e^{-16}, e^2]\}$;
- *gamma*: Uniform choice $\{0$, Log-Uniform distribution $[e^{-16}, e^2]\}$.

### A.1.2 CATBOOST

For easy re-implementation, the hyperparameter variable names followed the sklearn package settings.

- *learning_rate*: Log-Uniform distribution $[e^{-5}, 1]$;
- *random_strength*: Discrete uniform distribution $[1, 20]$;
- *one_hot_max_size*: Discrete uniform distribution $[0, 25]$;
- *l2_leaf_reg*: Log-Uniform distribution $[1, 10]$;
- *bagging_temperature*: Uniform distribution $[0, 1]$;
- *leaf_estimation_iterations*: Discrete uniform distribution $[1, 10]$.

---

[2]https://github.com/hyperopt/hyperopt
[3]https://scikit-learn.org/

### A.1.3 NODE

- *number of layers*: $\{2, 4, 8\}$;
- *total number of trees*: $\{1024, 2048\}$;
- *tree depth*: $\{6, 8\}$;
- *output dimension of trees*: $\{2, 3\}$;
- *learning rate*: $10^{-3}$.

### A.1.4 NET-DNF

Following the original setting in (Abutbul et al., 2021), the learning rate is initialized as 0.05 and is reduced by monitoring the training loss (check its official codes for details).

- *number of formulas*: $\{64, 128, 256, 512, 1024, 2048, 3072\}$;
- *feature selection beta*: $\{1.6, 1.3, 1.0, 0.7, 0.4, 0.1\}$.

### A.1.5 TABNET

- $\lambda_{sparse}$: $\{0.0, 10^{-6}, 10^{-4}, 10^{-3}, 10^{-2}, 0.1\}$;
- $\gamma$: $\{1.0, 1.2, 1.5, 2.0\}$;
- *batch size $B$*: $\{256, 512, 1024, 2048, 4096, 8192, 16384, 32768\}$;
- $N_d$ *and* $N_a$: $\{8, 16, 24, 32, 64, 128\}$;
- $N_{steps}$: $\{3, 4, 5, 6, 7, 8, 9, 10\}$;
- $B_V$: $\{256, 512, 1024, 2048, 4096\}$;
- $m_B$: $\{0.6, 0.7, 0.8, 0.9, 0.95, 0.98\}$;
- *learning rate*: $\{0.002, 0.01, 0.02, 0.025\}$.

### A.1.6 FCNN

- *number of layers*: Discrete uniform distribution $[2, 9]$;
- *output dimensions of full-connections*: Discrete uniform distribution $\{128, 256, 512, 1024\}$;
- *learning rate*: Uniform distribution $[10^{-4}, 10^{-2}]$;
- *dropout rate*: Uniform distribution $[0, 0.5]$;
- *lasso coefficient (optional)*: Discrete uniform distribution uniform $[10^{-7}, 10^{-4}]$;
- *learning rate*: Uniform distribution $[10^{-4}, 0.01]$.

### A.1.7 FT-TRANSFORMER

- *the number of layers*: Discrete uniform distribution $[1, 4]$;
- *feature embedding size*: Discrete uniform distribution $[64, 512]$;
- *residual dropout*: Uniform distribution $[0, 0.2]$;
- *attention dropout*: Uniform distribution $[0, 0.5]$;
- *FFN dropout*: Uniform distribution $[0, 0.5]$;
- *FFN factor*: Uniform distribution $[\frac{2}{3}, \frac{8}{3}]$;
- *learning rate*: Log-Uniform distribution $[10^{-5}, 10^{-3}]$;
- *weight decay*: Log-Uniform distribution $[10^{-6}, 10^{-3}]$;
- *the number of iterations*: 100.

### A.1.8 DANET-24

The learning rate of DANet-24 is initially set 0.008 and is decayed by 5% in every 20 epochs.

- $k_0$: $\{5, 8\}$;
- $d_0$: $\{32, 48\}$;
- $d_1$: $\{64, 96\}$.

### A.1.9 VECTOR CAPSNET

- *learning rate*: Uniform distribution [0.02, 0.1];
- *primary capsule size*: Discrete uniform distribution [64, 128];
- *the number of primary capsules*: Discrete uniform distribution [4, 32];
- *senior capsule size*: Discrete uniform distribution [4, 32].

### A.1.10 TABCAPS

- *learning rate*: Uniform distribution [0.02, 0.1];
- *the number of senior capsules for a class*: Discrete uniform distribution [1, 5];
- *primary capsule size*: Discrete uniform distribution [64, 128];
- *the number of primary capsules*: Discrete uniform distribution [4, 32];
- *senior capsule size*: Discrete uniform distribution [4, 32];
- *the number of learnable "words"*: Discrete uniform distribution [16, 64].

## A.2 Classification Performances on "Accuracy" and "ROC-AUC"

Here we report the model performances measured by the "accuracy" in Table 4 and the binary classification performances measured by "ROC-AUC" in Table 5. These classification performances are obtained in 5 runs with regard to random seeds. One can see that our TABCAPS obtains better or competitive performances in comparison with previous works.

Table 4: **Classification Performances measured by "accuracy".**

| Method | Click | Diabetes | EEG | Gas | Heart | Hill | Hig | Epsilon |
|---|---|---|---|---|---|---|---|---|
| XGboost | 0.6666 | 0.9575 | 0.9342 | 0.9957 | 0.8833 | 0.5761 | 0.7262 | 0.8884 |
| Catboost | 0.6599 | 0.9250 | 0.9322 | 0.9935 | 0.8501 | 0.4856 | 0.7256 | 0.8878 |
| TabNet | 0.6684 | 0.9725 | 0.8815 | 0.9949 | 0.8166 | 0.9012 | 0.7190 | 0.8890 |
| Net-DNF | 0.6694 | 0.9775 | 0.9379 | 0.9942 | 0.8333 | 0.7819 | 0.7268 | 0.8922 |
| NODE | 0.6688 | 0.7903 | 0.7871 | 0.8606 | 0.8667 | 0.6214 | 0.7260 | 0.8952 |
| FT-Transformer | 0.6606 | 0.9725 | 0.9826 | 0.9953 | 0.9166 | 0.8971 | 0.7290 | 0.8976 |
| DANet-24 | 0.6698 | 0.9687 | 0.9776 | 0.9949 | 0.8333 | 0.9095 | 0.7284 | 0.8962 |
| FCNN mixup | 0.6674 | 0.9375 | 0.9746 | 0.9956 | 0.7333 | 0.9136 | 0.7075 | 0.8934 |
| FCNN lasso | 0.6670 | 0.9403 | 0.9639 | 0.9863 | 0.6677 | 0.8949 | 0.6677 | 0.8958 |
| Vector CapsNet | 0.6678 | 0.9025 | 0.9726 | 0.9939 | 0.6833 | 0.6419 | 0.7102 | 0.8905 |
| TABCAPS (Ours) | 0.6685 | 0.9725 | 0.9776 | 0.9968 | 0.9023 | 0.9012 | 0.7300 | 0.8975 |

Table 5: **Binary classification Performances measured by "ROC-AUC".** The *Gas* dataset is not included because it is not for binary classification.

| Method | Click | Diabetes | EEG | Heart | Hill | Hig | Epsilon |
|---|---|---|---|---|---|---|---|
| XGboost | 0.7216 | 0.9859 | 0.9884 | 0.9447 | 0.5939 | 0.8054 | 0.9542 |
| Catboost | 0.7199 | 0.9878 | 0.9859 | 0.9330 | 0.4965 | 0.8048 | 0.9524 |
| TabNet | 0.7122 | 0.9853 | 0.9535 | 0.9095 | 0.9632 | 0.7911 | 0.9562 |
| Net-DNF | 0.7169 | 0.9932 | 0.9838 | 0.9412 | 0.8707 | 0.8018 | 0.9532 |
| NODE | 0.7146 | 0.8411 | 0.8173 | 0.9506 | 0.6626 | 0.7444 | 0.9497 |
| FT-Transformer | 0.7180 | 0.9883 | 0.9978 | 0.9764 | 0.9833 | 0.8078 | 0.9586 |
| DANet-24 | 0.7189 | 0.9825 | 0.9981 | 0.8931 | 0.9710 | 0.8042 | 0.9582 |
| FCNN mixup | 0.7178 | 0.9653 | 0.9962 | 0.8978 | 0.9531 | 0.7844 | 0.9598 |
| FCNN lasso | 0.7128 | 0.9702 | 0.9958 | 0.8343 | 0.5809 | 0.7247 | 0.9323 |
| Vector CapsNet | 0.7168 | 0.9902 | 0.9965 | 0.9036 | 0.5724 | 0.7754 | 0.9545 |
| TABCAPS (Ours) | 0.7199 | 0.9907 | 0.9970 | 0.9436 | 0.9911 | 0.8081 | 0.9606 |

## A.3 Performances with other Embedding Approaches

In this paper, we devise multivariate Gaussian kernels to synthesize vectorial features. Here we compare our proposed multivariate Gaussian embedding approach with the commonly used linear feature embedding approach (like the embedding approach in FT-Transformer). On the *Diabetes* dataset, TABCAPS with the linear embedding layer achieves 0.1328 on log-loss ($\downarrow$) while TABCAPS with multivariate Gaussian kernels achieves 0.1204. The comparison results suggest that our multivariate Gaussian kernels is beneficial, which might be due to the non-linearity of the multivariate Gaussian embedding approach.

