# OpenReview forum: "TabCaps: A Capsule Neural Network for Tabular Data Classification with BoW Routing"
_ICLR.cc/2023/Conference — ICLR 2023 poster_

### Official Review · Reviewer_sBG9 · 2022-10-22

**Confidence:** 2
**Clarity, Quality, Novelty And Reproducibility:** The clarity, quality novelty, and rep…
**Correctness:** 3
**Technical Novelty And Significance:** 2
**Empirical Novelty And Significance:** 2
**Recommendation:** 6

**Details Of Ethics Concerns:**

No concern.

**Strength And Weaknesses:**

Strengths:

1. The idea of using capsules to encapsulate all tabular features of an instance into vectorial features is interesting. In this way, we do not have to deal with individual features so we may integrate heterogeneous features in learning.
2. TabCaps achieves good performance with a small number of parameters in the six datasets.
3. TabCaps also shows strong generalization performance.

Weaknesses:

1. It is counter-intuitive to me why deep learning methods have not encapsulated all tabular features. With several layers of networks and by embedding them into the hidden representations, deep nets could also learn how to integrate the information of individual features. That being said, it is unclear why TabCaps can achieve better performance.
2. Only six tabular datasets are considered. Experiments on more datasets will enhance the results.

**Summary Of The Paper:**

This paper proposes a method to encode the tabular features into vectorial space. To this end, the paper devises a capsule network to learn from the vectorial features by Gaussian kernels and routing method. Experiments on six real-world tabular datasets are conducted to validate the effectiveness of the proposed method named TabCaps. The results suggest that TabCaps outperforms or is competitive with the previous deep learning approaches.

**Summary Of The Review:**

This work proposes a capsule-based framework to learn from tabular data. The idea is interesting, but it is unclear why the method can perform well and what the advantages are over the existing deep learning methods. The experiments are not convincing as well since there are only six datasets.

---

> ### Author Response · Authors · 2022-11-18
> **To Reviewer sBG9**
>
> Thank you very much for finding our idea interesting. We hope our responses can address your concerns.
>
> **(1) Why deep learning methods have not encapsulated all tabular features? Why does TabCaps perform well?**
>
> Thank you for the insightful question.
>
> In previous deep learning methods (e.g., DANets [1], TabNet [2], and FT-Transformer [3]), each tabular feature value is regarded as an independent element. An obvious example is MLP, which makes all the tabular features interact with each other. However, tabular data are heterogeneous, and thus some inappropriate feature interactions might lead to inferior performances.
>
> Previous works tried to use some sparse filters[1]/ attention approaches [2,3] to find relevant tabular features for better interactions. Empirically, we found such approaches were somewhat beneficial, but still introduced some unwanted feature interactions. A direct evidence is the Table 3 of DANets [1]. One can see that, although the relative features are found, some unwanted feature interactions (the light-colored parts) were also used. An ideal approach should not only find all relative features but also exclude all of the irrelative features. However, in a relation-agnostic situation, it is hard to conduct all the feature interactions in the right way (as stated in the "Related Work").
>
> Different from previous work, in our TabCaps, we encapsulated all tabular features and avoids the feature interactions directly. It is a novel paradigm and we empirically proved that it can achieve better performances, which may inspire future work on tabular data. Our approach proves that reducing the "wrong" feature interactions is just as important, and sometimes more important, than performing the right ones.
>
> **(2) More experimental results**
>
> Thank you for constructive suggestion! We further tested our approaches on two commonly used datasets Higgs-small and Epsilon, and obtained good performances: 0.73004 and 0.8987 on accuracy ($\uparrow$), outperforming the performances of previous SOTA FT-Transformer (0.729 and 0.8982).
>
> Since the performances of all the compared methods on log-loss further requires hyper-parameter tuning, we report the performances of accuracy here and we will add the performances on log-loss to Table 1 in the final version. Thank you!
>
> **References:**
>
> [1] DANets: Deep abstract networks for tabular data classification and regression. In AAAI, 2022.
>
> [2] TabNet: Attentive interpretable tabular learning. In AAAI, 2021.
>
> [3] Revisiting deep learning models for tabular data. In NeurIPS. 2021.

---

> > ### Comment · Reviewer_sBG9 · 2022-11-21
> > **Thank you for the reponse**
> >
> > Thank you for running additional experiments. It would be great if the authors can update the accuracy results in Table 4 for a better presentation of the results.

---

> > > ### Author Response · Authors · 2022-11-21
> > > **To Reviewer sBG9**
> > >
> > > Thank you for the careful suggestion! We will update the results of the additional datasets Higgs-small and Epsilon into Table 1, 2, 4. The performances of TabCaps on accuracy surpass previous SOTA FT-Transformer, which verified the superiority of our method.

---

> ### Author Response · Authors · 2022-11-25
> **To Reviewer sBG9**
>
> We would like to update the performances on two datasets, Epsilon and Higgs-small. Our TabCaps outperformed previous works on Epsilon (Log-loss), Epsilon (Accuracy), and Higgs-small ( Accuracy), and obtained competitive result on Higgs-small with the Log-loss metric. Since our TabCaps obtained considerable performances with no feature interactions (Other DNN approaches conduct feature iteraction), it might inspire the future work on rethinking the necessity and design of feature interaction approaches. We will update the results of tables in the final version. Thank you!
>
> | Model                                                 | Epsilon (Log-loss, $\downarrow$)    | Epsilon (Accuracy, $\uparrow$) | Higgs-small (Log-loss, $\downarrow$) | Higgs-small ( Accuracy, $\uparrow$) |
> | :-----------------------------------------------  | :------- | :------- | :------- |:------- |
> | TabNet                                                |    0.2631 |  0.8896  |  0.5467 |  0.7190 |
> | Net-DNF                                             |    0.2547  |  0.8932 |  0.5342 |  0.7268 |
> | NODE                                                 |   0.2633 |  0.8958 |  0.6186 | 0.7260 |
> | FT-T                                                   |   0.2539   |  0.8982 |  0.5331 |  0.7290 |
> |DANet-24                                            |   0.2611 |  0.8962  |  **0.5303** | 0.7284 |
> |FCNN mixup                                       |  0.2634  |   0.8924  |  0.5679  | 0.7075 |
> |FCNN lasso                                        |  0.2994  |   0.8949  |  1.3210  | 0.6677 |
> |Vector CapsNet                                  |  0.5425  |  0.8922   |  0.6265 |  0.7102 |
> |TabCaps                                             |   **0.2510**  |  **0.8987** |  0.5378  |  **0.7300** |
>
> --------------------------------------------------------------------------------------------
> Note: For Epsilon, here we use 20% training data for validation. We use 15% (the same to other dataset) in the final version, so that the results are slightly different.

---

> > ### Comment · Reviewer_sBG9 · 2022-12-05
> > **Thank you for the additional results**
> >
> > Thank you for the additional results, and sorry for the late response. I will raise my score.

---

### Official Review · Reviewer_NLd7 · 2022-10-24

**Confidence:** 2
**Correctness:** 3
**Technical Novelty And Significance:** 2
**Empirical Novelty And Significance:** 2
**Recommendation:** 5

**Clarity, Quality, Novelty And Reproducibility:**

Clarity:

1. As discussed before, more motivation, intuition, or analysis are needed for the proposed method.

2. Introduction to the background of capsule networks is needed.

Quality:

1. It seems that different recent methods for tabular data use different datasets and evaluation metrics (e.g. log-loss, RMSE, accuracy). I was wondering how these settings are selected in this paper. Due to this, I feel it's hard to cross compare different methods in unified settings.

**Strength And Weaknesses:**

Strength:

1. I'm not an expert with capsule networks, but after some search, I didn't find the applications of capsule networks for tabular data. This paper might be the first one of applying it in this domain.

2. The paper provides a relatively comprehensive experiments with several advanced methods for tabular data.

Weaknesses:

1. The main contribution of the paper is making capsule networks work for tabular data by introducing a few adaptations and modifications, such as sparse projection and BOW routing. However, many of the proposed adaptations are lack of clear motivations or intuitions. For example, why BOW routing works particularly with tabular data? The paper shows how but misses why, which make some of the adaptations more like heuristics instead of principled approaches.

2. As the paper introduces several adaptations, although with the ablation study, I feel that more ablation study is needed to verify the performance of different combinations of the adaptations.

3. The performance improvement of the proposed method seems to be marginal in many cases, shown in Table 1, given the error bars, not mentioning that improvement of classification accuracy is hardly observed in Table 4.

**Summary Of The Paper:**

This paper presents a capsule network method for tabular data classification. To tailor the capsule network for this task, the paper introduces several configurations with the focus on so-called BoW routing.

**Summary Of The Review:**

The paper proposes a capsule network for tabular data, some adaptations of which seem a bit ad-hoc. The performance improvement seems a bit marginal.

---

> ### Author Response · Authors · 2022-11-18
> **To Reviewer NLd7**
>
> Thank you for your careful comments. We hope our responses can address your concerns.
>
> **(1) Motivations and intuitions of the proposed designs.**
>
> *(i) As for the BoW routing:*
>
> Previous CapsNets for images were proposed to learn the hierarchical concepts of objects (from part to whole). The routing algorithms were designed to group primary capsule features (representing object parts) to yield the senior capsule features (representing object wholes), and the feature grouping were typically conducted by some clustering-like process.
>
> To measure the feature similarity in grouping primary capsule features, an intuitive method is to directly compute the similarity between primary capsule features and senior capsule features in the original feature space. The dynamic routing is an approach that is based on such direct similarity measure, whose performances are inferior, see Table 1 and Table 3 (2). We consider that, the feature space highly related to the prediction target should be more beneficial in grouping desired features for better classification performance. Inspired by the BoW model that performs information retrieval in the semantic space guided by the "words", we implement a learnable BoW routing for our case. We added some statements (see page 4) to make it clear (in the **rebuttal version**). Thank you!
>
> *(ii) As for the sparse projection:*
>
> A projection is commonly used in previous CapsNets (as stated in the "Related Work" section). For tabular data, some features are irrelevant to the target [2], which should be ignored. Thus, we used a sparse projection for feature selection. Similar operations were used in previous tabular learning models, for example, TabNet [1] and DANets [2]. We tried to make it more clear in the **rebuttal version**.
>
> Thank you for your constructive suggestions.
>
> **(2) Cross comparison and performance gains.**
>
> We compared the performances of methods by running the official codes with the given hyper-parameter tuning settings, as stated in the "Experiments" section and the appendix. The data splits are also the same.
>
> Our method obtained the best or the second best performances on log-loss (Table 1 and 3), and the performances on accuracy is also better or competitive (Table 4), with remarkably lower computational cost.
>
> **However, please note that the major contribution of this paper is to use vectorial feature format for depicting the global semantics of tabular instances directly, which avoids the troublesome feature interactions.** Namely, this paper provides a new paradigm for tabular learning to design approach to directly learn instance-level semantics, which will inspire future work. The advantages of our design include the good performances (Table 1), good generalization capability (Table 2), user-friendly training phase without intensely monitoring the performances (Sec.4.4, Fig.3), low computational cost (Table 1). Thank you!
>
> **(3) More ablation studies.**
>
> Thank you for your suggestions. We added the two ablation studies to evaluate our designs (see Table 3 in the **rebuttal version**). (i) We reported the performances of "BoW routing + simple projection (without entmax)" and "dynamic routing + sparse projection". We found that both BoW routing and sparse projection are beneficial and are compatible with each other. (ii) We also reported the performances of TabCaps with different amounts of capsule layers, which showed that deeper model can perform slightly better, but a 2-layer model is sufficient to attain good performances.
>
> **(4) Metrics and datasets.**
>
> *(i) About the metrics:*
>
> In the field of tabular data learning, accuracy and log-loss are used in classification tasks, while MSE is used for regression. We report the performances on both log-loss and accuracy. We consider log-loss is widely used in various Kaggle competitions (e.g., the Otto competition https://www.kaggle.com/c/otto-group-product-classification-challenge/overview/evaluation) and thus we used it in the main paper (in Table 1).
>
> *(ii) About the used datasets:*
>
> Notably, no “standard” datasets are known in the tabular learning area because tabular features are of different distributions, which is different from image area in which features follow similar distributions. For fair comparison, we selected those datasets with "few or no categorical features, eliminating the impacts of various feature embedding approaches for categorical features in model comparison.", as stated in page 6.
>
> **References:**
>
> [1] TabNet: Attentive interpretable tabular learning. In AAAI, 2021.
>
> [2] DANets: Deep abstract networks for tabular data classification and regression. In AAAI, 2022.

---

> > ### Author Response · Authors · 2022-11-18
> > **To Reviewer NLd7**
> >
> > **(5) Background of CapsNets.**
> >
> > Thank you for your helpful suggestion! We agree that the background of CapsNets should be more clear, and we reorganized the related work to highlight the basic concepts of CapsNets, and added more details about operations (see the **rebuttal version**). Besides, we also move the "related work" section ahead of the "method" section to make this paper easier to understand.

---

### Official Review · Reviewer_NQgm · 2022-10-24

**Confidence:** 4
**Correctness:** 4
**Technical Novelty And Significance:** 3
**Empirical Novelty And Significance:** 3
**Recommendation:** 8

**Clarity, Quality, Novelty And Reproducibility:**

The paper is well written and easy to follow but assumes some prior knowledge of capsnets. There is some quality work presented here and the appendix helps to get an idea of hyperparameters used - releasing the code post-acceptance would be helpful so that people can reproduce it more easily.

A major limitation is the size of the dataset - It would have been nice to compare (or reflect upon) performance on larger datasets, which is an area CapsNet have been suffering from in general. Some additional visualisations on how the learning behaves from one layer to another would have been helpful - I know first-hand that this is not as straightforward as it is with CNNs, but it would be nice to show what the capsules learn, especially during routing/voting. It is more intuitive for images, but how does this actually work with tabular data?



**Strength And Weaknesses:**

Strengths:
There is a good empirical work presented in this paper.
a) It makes a decent step forward on capsnet research b) The experimental setup and datasets considered are adequate c) the method appears well-grounded and sound and the authors have provided an appendix with the hyperparameters.

Weaknesses: a) I think the paper is positioned with respect to the original methods of Sabour and Hinton (dynamic routing and EM respectively) - that is not correct in my opinion as most of the works that followed up from these have gone on improving pretty much every facet of those algorithms, in terms of performance, number of parameters, iterative vs non-iterative, etc. some of these works have already been cited some others not, but the points made in the paper would be stronger had the authors focused on presenting the improvements with respect to the current state of the art (not in terms of results). b) I think some reflection on how the approach scales would have been helpful - for instance, traditional capsnet suffer from the collapse of learning making it hard to train, say on imagenet, or stuck several capsule layers.



**Summary Of The Paper:**

This paper proposes a bespoke CapsNet model called TabCaps that is tailored to tabular data processing. This seems to be the first effort to develop a bottom-up approach specifically for tabular data whereby all features are considered as a vector (per the properties of capsules) which are then transformed by a multivariate Gaussian kernel with some learnable parameters so that they can be adapted to the dataset in question. The higher-level capsules (called senior capsules) are tasked to learn the semantics of the target classes. This is achieved through a feature transformation by a sparse weight matrix into feature votes which are then fed into a newly proposed "differentiable" Bag of Words. The vector lengths of the senior capsules represent the existence probabilities of a specific semantic, whereby each class is represented by a number of senior capsules, akin to ensembles (authors use more than one senior capsule hence the ensemble, as opposed to previous work). The results are quite significant and impressive in that they are outperforming the defacto approaches that are based on XGBoost across several benchmark datasets.



**Summary Of The Review:**

I personally believe that CapsNets are powerful learners that have also recently been shown to relate to the attention mechanism popularised by transformers (https://arxiv.org/pdf/2206.02664.pdf). Conceptually they are better grounded and more explainable. This paper contributes by presenting an alternative view of capsnets this time on tabular data that as a researcher myself would like to see published to support further fundamental research on capsnets.

---

> ### Author Response · Authors · 2022-11-18
> **To Reviewer NQgm**
>
> Thank you very much for recognizing our paper as a decent step forward on CapsNet research. We hope our responses can address your concerns.
>
> **(1) Analysis and comparison with SOTA CapsNets, and why BoW Routing is efficient**
>
> Thank you for your careful suggestion. In previous work for image processing, "a primary capsule may capture any object parts and thus the initialized senior capsule features are unknown before the routing step'', as stated in page 4. Since most of previous CapsNets were proposed for images and tried to merge object parts by the “routing-by-agreement” process to represent various object wholes, the routing algorithms were typically implemented by some clustering-like approaches with iterations (e.g., dynamic routing, EM routing). Some recent routing algorithms avoided iterations by some delicate designs (e.g., using agglomerative hierarchical clustering [1] or variational bayes approach [2]), which did not change the function of routing algorithm that performs a clustering-like process.
>
> However, in our approach on tabular data, “each primary capsule in our design learns concrete semantics that represent the marginal likelihoods of a specific multivariate Gaussian distribution (i.e., a specific instance type)” (stated in page 4). Different from those SOTA non-iterative routing algorithms in which the clustering-like processes were performed straightforwardly by some delicate designs, our BoW routing performing in a straightforward way “partially results from the innovative definition of the primary capsules that learns concrete semantics, allowing to predict initialized senior capsule features directly” (stated in page 5).
> Our designs (our TabCaps and routing algorithm) were proved to be effective on tabular data by the experiment in Sec.4.2. The results showed that our BoW routing finds the clear paths to yield senior capsule features, which is stable in dealing with different samples (with the std of the transformation weights in 0.001-0.003).
> Namely, our method is natively different from previous routing algorithms for images due to the way to learn the primary capsule features, including not only the original methods of Sabour and Hinton, but also the recently proposed methods [1, 2]. We consider that this design might be hard to be utilized on the image tasks, because the tabular data are structured data but images are not.
>
> We modified some statements in the method and experiments to make it clear in the **rebuttal version**. Thank you.
>
> **(2) The reflections on how the approach scales (with more capsule layers).**
>
> Thank you for your constructive suggestion. In our design, our routing algorithm naturally performs non-iteration process, and thus our TabCaps can go deeper (a direct evidence is that DeepCaps [3] removed the iteration and stacked several layers). The reason why our RoW routing can perform straightforwardly is discussed in **(1)**. Empirically, we found the performances of models with 2 layers, 3 layers, and 4 layers were competitive. We added the ablation studies on the number of capsule layer (see Table 3) and the discussions (see page 9). The TabCaps with 2, 3, 4 capsule layers obtain 0.12043, 0.06254, 0.11138 on the Diabetes dataset, and obtain 0.34047, 0.33992, and 0.33673 on the Heart dataset. A 2-layer model is sufficient to obtain good performances on various datasets. To balance the performances and efficiency, we used 2 capsule layers as the default setting.
>
> **(3) The size of datasets.**
>
> In our experiments, we tested our CapsNets on some commonly used datasets, some of which are large.
>
> (i) The dataset Hill and Gas have 101 and 128 tabular features, respectively.
>
> (ii) Click dataset contains 1 million samples, which is competitive with ImageNet.
>
> We considered that previous CapsNets obtained inferior performances on images, because of the insufficient capsule numbers to parse complex semantics in images. However, since tabular data are structured data, the semantics are separated and more clear.
>
> In addition, we add experiments on Higgs-small (100K data) and Epsilon (500K data and 2000 features). These datasets contain many samples or have many tabular features. Since we have not finished all of the hyper-parameter tuning for the compared methods on Log-loss, we compare the performances on accuracy in reference to the performances reported in the previous SOTA model FT-T paper [4]. The accuracy of TabCaps on Higgs-small is 0.73004 ($\uparrow$), better than previous SOTA FT-T 0.729. While the accuracy of TabCaps on Epsilon is 0.8987 ($\uparrow$), better than previous SOTA FT-T 0.8982. We will add the performances on log-loss to the final version. Thank you!
>
>
> **References:**
>
> [1] A Receptor Skeleton for Capsule Neural Networks. In ICML, 2021.
>
> [2] Capsule routing via variational bayes. In AAAI, 2020.
>
> [3] Deepcaps: Going deeper with capsule networks. In CVPR, 2019.
>
> [4] Revisiting deep learning models for tabular data. In NeurIPS. 2021.

---

> > ### Author Response · Authors · 2022-11-18
> > **To Reviewer NQgm**
> >
> > **(4) The learning behavior of our CapsNet.**
> >
> > As discussed in **(1)**, the primary capsules in our model learns concrete semantics so that our BoW routing can find a clear path to yield the senior capsule features. The learning behavior of TabCaps is visualized in Fig.4 and is discussed in Sec.4.5. We found the transformation weight maps are sparse and transformation weights are stable (the standard deviation of the weights are in 0.01-0.03). This finding is expected, and indicates that our routing actually captures some stable relations between different "tabular instance types" (depicted by multiple Gaussian kernels), which verifies a non-iteractive routing process is beneficial.

---

> > > ### Comment · Reviewer_NQgm · 2022-11-21
> > > **Adequately addressed**
> > >
> > > Thanks to the authors for providing detailed responses to my comments! Please make sure to add all the above to your final version, including the newly identified references, should your paper be accepted.

---

> > > > ### Author Response · Authors · 2022-11-21
> > > > **To Reviewer NQgm**
> > > >
> > > > Thanks to Reviewer NQgm for the constructive suggestions. We'll definitely add all the above to the final version.

---

> ### Author Response · Authors · 2022-11-25
> **To Reviewer NQgm**
>
> We would like to update the performances on two large-scale datasets, Epsilon and Higgs-small. Our TabCaps outperformed previous works on Epsilon (Log-loss), Epsilon (Accuracy), and Higgs-small ( Accuracy), and obtained competitive result on Higgs-small with the Log-loss metric. We will add all the results and the newly identified references in the final version. Thank you!
>
> | Model                                                 | Epsilon (Log-loss, $\downarrow$)    | Epsilon (Accuracy, $\uparrow$) | Higgs-small (Log-loss, $\downarrow$) | Higgs-small ( Accuracy, $\uparrow$) |
> | :-----------------------------------------------  | :------- | :------- | :------- |:------- |
> | TabNet                                                |    0.2631 |  0.8896  |  0.5467 |  0.7190 |
> | Net-DNF                                             |    0.2547  |  0.8932 |  0.5342 |  0.7268 |
> | NODE                                                 |  0.2633   |  0.8958 |  0.6186 | 0.7260 |
> | FT-T                                                   |   0.2539   |  0.8982 |  0.5331 |  0.7290 |
> |DANet-24                                            |   0.2611 |  0.8962  |  **0.5303** | 0.7284 |
> |FCNN mixup                                       |  0.2634  |   0.8924  |  0.5679  | 0.7075 |
> |FCNN lasso                                        |  0.2994  |   0.8949  |  1.3210  | 0.6677 |
> |Vector CapsNet                                  |  0.5425  |  0.8922   |  0.6265 |  0.7102 |
> |TabCaps                                             |   **0.2510**  |  **0.8987** |  0.5378  |  **0.7300** |
>
> ------------------------------------
> Note: For Epsilon, here we use 20% training data for validation. We use 15% (the same to other dataset) in the final version, so that the results are slightly different.

---

### Official Review · Reviewer_Zv1Z · 2022-11-29

**Confidence:** 4
**Correctness:** 3
**Technical Novelty And Significance:** 3
**Empirical Novelty And Significance:** 2
**Recommendation:** 6

**Clarity, Quality, Novelty And Reproducibility:**

Clarity: The paper is well-written and easy to follow.

Quality: The proposed method technologically sounds good, and the experiment design is solid.

Novelty: I think it is the first work to use Capsule Neural Network in tabular data.

Reproducibility: The details in the appendix help to reproduce, but it would be better if the authors can release the source code.

**Strength And Weaknesses:**

Strengths

- The paper is well written.
- The experiment setting is comprehensive.
- The idea is interesting, and the experiment results demonstrate looks good.


Weaknesses  & Questions

1. Current method seems cannot apply to regression tasks.
2. The experiment result seems not very significant, especially the accuracy comparison from Table 4.
3. Can you explicitly show #classes of different tasks in the experiments? And I would like to see the AUC metric for the binary classification, rather than log-loss. Besides, how do you use XGBoost for multi-class (K > 2) tasks?
4. Can we replace the "Multivariate Gaussian Kernels" with FFN? If it can, how about the performance? if not, why?
5. In the widely used boosting tree based methods, there are many interactions among tabular features. It surprised me that non-feature-interaction is better in neural networks. Can authors explain more about that?

**Summary Of The Paper:**

This paper proposed TabCaps, to process the multiple features into one vectorized unit, avoiding the interaction among tabular features.  In TabCaps, a tabular instance is respectively encoded into several vectorial features by some optimizable multivariate Gaussian kernels in the primary capsule layer, where each vectorial feature represents a specific "profile" of the input instance and is transformed into senior capsule layer under the guidance of a novel straightforward routing algorithm. Rather than clustering, an efficient bag-of-words is used in routing.


**Summary Of The Review:**

Overall, I lean to accept the paper. Although the experiment result is not very significant, I think this is a good paper, since it may inspire future research in neural networks for tabular data.

---

> ### Author Response · Authors · 2022-12-06
> **To Reviewer Zv1Z**
>
> Thank you very much for finding our idea interesting. We hope our responses can address all your concerns.
>
> **(1) Current model can only for classification tasks.**
>
> Our approach is based on the capsule neural network (CapsNet), a special kind of neural network for classification tasks. Since how to use CapsNets on regression tasks is another research topic, this paper only focuses on exploring non-feature-interaction approach on tabular data. Our work verifies that non-feature-interaction approach can also achieve considerable performances, which may inspire further work to explore the direct instance-level semantic learning (as stated in Sec. Conclusions).
>
> **(2) The accuracy in Table 4.**
>
> The accuracy on tabular data tasks is somewhat saturated, and it is widely recommended to use log-loss (e.g., the Kaggle competition https://www.kaggle.com/c/otto-group-product-classification-challenge/overview/evaluation), a rigorous metric that is the maximum likelihood function to depict the degree to which the model fits the test data. The log-loss results show that our method fit test data much better than the compared methods, suggesting that the feasibility of our approach and inspiring future work to explore instance-level semantic learning practice without feature-wise interactions.
>
> **(3) How to use XGBoost on multi-class tasks?**
>
> It is available to perform XGBoost on multi-class tasks by using "merror" loss function (merror = multiclass classification error rate), as shown in https://xgboost.readthedocs.io/en/stable/python/examples/custom_softmax.html. Some previous tabular learning works also used XGBoost in the same way. Thank you!
>
> **(4) Replace "multivariate Gaussian kernels" with "FFN"?**
>
> Thanks for the insightful question. We consider there are two kinds of "FFN" that can be used to replace our proposed "multivariate Gaussian kernels". **(i)** Fully-connected layers that conduct fully feature interactions to take place of the multivariate Gaussian kernels. The model with fully-connected layer obtains 0.1372 (log-loss, $\downarrow$) on the Diabetes dataset (following the ablation study), which is worse than the counterpart with our multivariate-Gaussian-kernel layer (0.1204). **(ii)** An embedding layer that employs a learnable vector to point-wise multiply with the input features (without feature interactions). The model with the embedding layer achieves 0.1328 on log-loss ($\downarrow$), worse than using the multivariate-Gaussian-kernel layer (0.1204) but better than using the fully-connected layer (0.1372). These results suggest that our multivariate Gaussian kernels is beneficial. We will add these results to the ablation study in our final version.
>
> **(5) Why can a non-feature-interaction approach work?**
>
> Thank you for the insightful question!
>
> Previous works often considered that the feature interactions are important in tabular learning practice. However, it is also witnessed that neural network approaches did not well capture the feature correlations. We noticed that previous neural networks for tabular data might capture some false positive feature correlations. A direct evidence is the visualization (Table 3) of DANets [1]. One can see that the model not only captures some "true" feature correlations that is considered to be beneficial, but also captures some "false positive" correlations that may hurt the performances. Ng [2] found that the neural network methods might be inefficient in processing data that has only a few relevant features. Namely, neural networks have been shown to be not good at processing tabular data, which has only a few relevant features. Therefore, a non-feature-interaction approach has a chance to obtain considerable performances.
>
> Actually, there are some classic data patterns (i.e., the "profile" mentioned in the paper) in the tabular data. Our proposed CapsNet learns the "profiles", which also implicitly model the tabular feature correlations (as stated in the penultimate paragraph in page 3).
>
> **References:**
>
> [1] DANets: Deep abstract networks for tabular data classification and regression. In AAAI, 2022.
>
> [2] Ng, Andrew Y. Feature selection, L 1 vs. L 2 regularization, and rotational invariance. In ICML. 2004.
>
> (*continued*)

---

> > ### Author Response · Authors · 2022-12-06
> > **To Reviewer Zv1Z**
> >
> > **(6) #. classes of different tasks and the AUC performances on binary-classification tasks.**
> >
> > Thank you. Except for the Gas dataset containing 6 classes, all the rest datasets have 2 classes. We will add the detailed dataset information to the final version.
> >
> > The performances on AUC are reported below and our TabCaps attained the best performances on 5 out of 7 datasets. We notice that the conclusions obtained from AUC performances are highly consistent to the conclusions obtained from log-loss, which shows the potential of our non-feature-interaction approach.
> >
> > |         | Click | Diabetes | EEG| Heart | Hill | Higgs | Epsilon |
> > | :----- | :------- | :------- | :------- | :------- |:------- | :------- | :------- |
> > | TabNet | 0.7122 | 0.9853 | 0.9535 | 0.9095 | 0.9632 | 0.7911 | 0.9581 |
> > | Net-DNF | 0.7169 | 0.9932 | 0.9838 | 0.9412 | 0.8707 | 0.8018 | 0.9579 |
> > | NODE | 0.7146 | 0.8411 | 0.8173 | 0.9506 | 0.6626 | 0.7444 | 0.9528 |
> > | FT-T | 0.7180 | 0.9883 | 0.9978 | **0.9764** | 0.9833 | 0.8078 | 0.9592 |
> > | DANet-24 | 0.7189 | 0.9825 | **0.9981** | 0.8931 | 0.9710 | 0.8042 | 0.9595 |
> > | Vector Capsule | 0.7168 | 0.9902 | 0.9965 | 0.9036 | 0.5724 | 0.7754 | 0.9549 |
> > | FCNN mixup | 0.7178 | 0.9653 | 0.9962 | 0.8978 | 0.9531 | 0.7844 | 0.9594 |
> > | FCNN lasso | 0.7128 | 0.9702 | 0.9958 | 0.8343 | 0.5809 | 0.7247 | 0.9448 |
> > | TabCaps (Ours) |  **0.7199** | **0.9907** | 0.9970 | 0.9436 | **0.9911** | **0.8081** | **0.9616**|
> >
> > ----------------------------------------------------
> > Note: For Epsilon, here we use 20% training data for validation. We use 15% (the same to other dataset) in the final version, so that the results are slightly different.

---

### Decision · Program_Chairs · 2023-01-20

**Decision:**

Accept: poster

**Justification For Why Not Higher Score:**

More empirical significance would be helpful.

**Justification For Why Not Lower Score:**

Good step towards making capsule nets work for tabular data.

**Metareview: Summary, Strengths And Weaknesses:**

The paper proposes a capsule neural network for tabular data classification.

The reviewers agreed the paper is well-written, with good experimental results and setup, and the method appears to be well-grounded in its motivation. The reviewers also agreed the paper is a nice step towards popularizing capsule networks in the context of tabular data.

Expanding the results to more datasets with clear gains would improve the significance of the paper.





**Note From Pc:**

if the above contains the word "oral" or "spotlight" please see: "oral" presentation means -> notable-top-5% and "spotlight" means -> notable-top-25%. As stated in our emails, we are disassociating presentation type from AC recommendations

**Summary Of Ac-Reviewer Meeting:**

Reviewers were unresponsive in general.

Some reviewers responded to request for availability for a virtual meeting. I held the meeting but none of them attended, although one reviewer (Zv1Z) notified me that he was sick with Covid and couldn't attend.